# C. elegans LIN-66 mediates EIF-3/eIF3-dependent protein translation via a cold-shock domain

Stephen M Blazie [iD], Daniel Fortunati [iD], Yan Zhao, Yishi Jin [iD]

**Protein translation initiation is a conserved process involving many proteins acting in concert. The 13 subunit eukaryotic initiation factor 3 (eIF3) complex is essential for assembly of the pre-initiation complex that scans mRNA and positions ribosome at the initiation codon. We previously reported that a gain-of-function (gf) mutation affecting the G subunit of the *Caenorhabditis elegans* eIF3 complex, *eif-3.g(gf)*, selectively modulates protein translation in the ventral cord cholinergic motor neurons. Here, through unbiased genetic suppressor screening, we identified that the gene *lin-66* mediates *eif-3.g(gf)*-dependent protein translation in motor neurons. LIN-66 is composed largely of low-complexity amino acid sequences with unknown functional domains. We combined bioinformatics analysis with in vivo functional dissection and identified a cold-shock domain in LIN-66 critical for its function. In cholinergic motor neurons, LIN-66 shows a close association with EIF-3.G in the cytoplasm. The low-complexity amino acid sequences of LIN-66 modulate its subcellular pattern. As cold-shock domains function broadly in RNA regulation, we propose that LIN-66 mediates stimulus-dependent protein translation by facilitating the interaction of mRNAs with EIF-3.G.**

## Introduction

Protein translation initiation in eukaryotic cells is highly regulated and involves coordinated actions of multiple protein complexes, known as eIF1-6 (Hinnebusch & Lorsch, 2012). The eIF complexes work in concert to orchestrate orderly assembly of ribosome subunits with mRNAs and an initiator tRNA. eIF3 is the largest translation initiation protein complex, consisting of 13 subunits, and recruits the small ribosomal subunit and mRNAs to form the 43S pre-initiation complex, which scans the 5′ untranslated region of mRNAs to locate the initiation codon (Cate, 2017). Consistent with their critical roles in protein translation, knockout or knockdown of many subunits of eIF3 generally results in inviable cells or organisms (Hanachi et al, 1999; Kamath et al, 2003; Zhuo

et al, 2023). However, recent studies have suggested that specific subunits of eIF3 can also provide precise temporal and spatial regulation of protein translation in cell proliferation, development, and stress response (Lee et al, 2015, 2016). Emerging mechanistic studies suggest that such eIF3 subunit–specific functions can be achieved through stimulus-induced protein phosphorylation and/or specific protein interaction partners (Lamper et al, 2020).

High-resolution cryo-electron microscopy and X-ray structures of the mammalian and yeast 43S pre-initiation complex have been reported (Hashem et al, 2013; Querol-Audi et al, 2013; Erzberger et al, 2014; des Georges et al, 2015). Eight subunits of eIF3 form a core module, and five subunits are located at the periphery of the 43S complex. The g subunit of eIF3 (eIF3g) resides in the periphery of the complex, binds the core subunit eIF3I through its N-terminal region, and interacts with RNA via an RNA recognition motif at the C-terminus (Hanachi et al, 1999). The middle region of eIF3g has a zinc finger (ZF) for which the structure has yet to be solved. Dysregulation of eIF3g has been implicated in human diseases. For example, the altered expression of eIF3g is associated with narcolepsy (Holm et al, 2015) and also observed in an animal model for autism (Hornberg et al, 2020). In Drosophila sensory neurons, knockdown of eIF3g impairs dendrite pruning (Rode et al, 2018).

We have previously reported a role of the *Caenorhabditis elegans* eIF3g subunit, EIF-3.G, in shaping the neuronal protein landscape in response to neuronal hyperexcitation (Blazie et al, 2021). A neuronal acetylcholine receptor ACR-2 is localized to the dendrites of the ventral cord cholinergic motor neurons, and a gain-of-function mutation in *acr-2* results in a leaky channel and causes hyperexcitation of the cholinergic motor neurons (Jospin et al, 2009). *acr-2(gf)* animals exhibit spontaneous whole-body convulsion. In genetic screening for behavioral suppression of *acr-2(gf)* mutants, we identified a missense mutation in the ZF of EIF-3.G (Blazie et al, 2021). Null mutants of *eif-3.G* are arrested in larval development. However, *eif-3.G* mutants that carry either the missense mutation or a small deletion of the ZF, designated *eif-3.G(gf)*, behave indistinguishably from WT in movement and health, but act cell-autonomously in the cholinergic motor neurons to ameliorate convulsion of *acr-2(gf)*. Using neuron-specific mRNA cross-linking, immunoprecipitation, and deep

Department of Neurobiology, School of Biological Sciences, University of California San Diego, La Jolla, CA, USA

Correspondence: yijin@ucsd.edu

sequencing, we have defined a mechanism by which EIF-3.G(GF) selectively modulates the translation of a set of mRNAs with GC-rich 5′ UTRs (untranslated regions) in the hyperexcited cholinergic motor neurons (Blazie et al, 2021).

Here, we report that the function of EIF-3.G(gf) in tuning neuronal protein translation involves the LIN-66 protein. We performed genetic screening to search for factors that reverted the behavioral suppression of *eif-3.G(gf)* on the convulsion of *acr-2(gf)* animals. We identified a loss-of-function mutation in *lin-66*, which was previously reported to be conserved in nematode species, with unknown functional domains (Morita & Han, 2006). Through informatics analysis of homologous protein structures, we identified that LIN-66 has a cold-shock domain embedded within low-complexity protein sequences. LIN-66 localizes to the somatic cytoplasm, closely associated with EIF-3.G. The localization pattern of LIN-66 depends on the low-complexity sequences. By functional dissection, we show that the cold-shock domain is critical for LIN-66 function. Our data uncover a previous unknown function of LIN-66 and provide in vivo insight into how EIF-3.G modulates neuron type–specific protein translation.

## Results

### Loss of function in *lin-66* suppresses the gain-of-function effect of *eif-3.G* on *acr-2(gf)* convulsion behavior

We aimed to identify genes that function in *eif-3.G(gf)*-dependent protein translation in hyperactivated cholinergic motor neurons of *acr-2(gf)* animals. We reasoned that genetic mutations reversing the suppression of *acr-2(gf)* convulsion by *eif-3.G(gf)* would likely reveal clues to such genes. We mutagenized *eif-3.G(gf); acr-2(gf)* double mutants, which exhibited wild-type–like locomotion. We screened F2 progeny for animals that displayed convulsions resembling *acr-2(gf)* single mutants (see the Materials and Methods section). After outcrossing analysis, we were able to re-isolate one mutation, *ju1661*, that exhibited specific suppression on *eif-3.G(gf)* in the context of *acr-2(gf)*. We then combined whole-genome sequencing analysis with single-nucleotide polymorphism (SNP) mapping of recombinants to localize *ju1661* to the right arm of chromosome IV. Examination of candidate SNPs that likely disrupt a gene's function revealed a point mutation changing the splice acceptor site of the third exon of the *lin-66* gene (Fig 1A).

The *lin-66* gene has seven exons and produces three mRNA isoforms (Fig 1A). The isoforms a and c differ by nine nucleotides as a result of alternative splice site usage at intron 3 and exon 4, and encode proteins of 627 aa and 624 aa, respectively. The isoform b has only exons 6 and 7, likely using alternative 5′ upstream sequences. To verify the effects of *ju1661* mutation, we isolated mRNAs from *ju1661* animals and obtained multiple *lin-66* cDNA fragments using primers annealing to exons 2 and 3. Sanger-sequencing analysis of these clones revealed both mis-spliced mRNA products using cryptic splice acceptor sites that would cause out-of-frame if translated, and correctly spliced mRNA products as observed in mature mRNAs of WT *lin-66*. Thus, *lin-66(ju1661)* causes

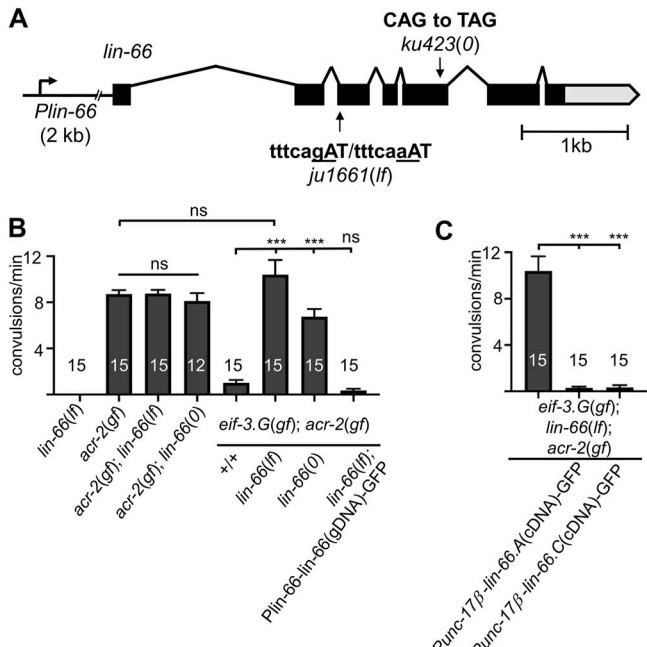

**Figure 1.** *lin-66* functions in cholinergic motor neurons to modulate *eif-3.g(gf)* activity.
**(A)** Illustration of the *lin-66* gene model on chromosome IV (modified from WormBase), along with the position and nucleotide change in *ju1661* and *ku423*, respectively. Black boxes represent exons; lines, introns; gray box, 3′ UTR; and the promoter, *Plin-66*, includes 2 kb upstream sequences. **(B, C)** Quantification of convulsion behavior in day 1 adult animals of the indicated genotypes. *lin-66(0)* animals were examined at L4 because of their larval arrest. Alleles used are as follows: *eif-3.g(ju807gf), acr-2(n2420gf), ju1661* as *lin-66(lf), ku423* as *lin-66(0)*. Convulsion quantification was from two to three independent observations, and at least two transgenic lines were scored. Statistics: one-way ANOVA with Bonferroni's post hoc test, *** P < 0.01 and ns, not significant.

partial defective pre-mRNA splicing and allows some production of wild-type proteins.

Null (0) mutations of *lin-66* were previously identified for their roles in temporal regulation of the vulva cell fate (Morita & Han, 2006). *lin-66(ku423)* is a nonsense mutation changing amino acid Gln378, and causes developmental arrest at late-stage larvae (L4 stage) (Fig 1A). In contrast, *lin-66(ju1661)* animals developed into fertile adults and displayed no discernable abnormalities in body shape, movement, and growth rate, consistent with the conclusion that *lin-66(ju1661)* retains partial function, designated as *lin-66(lf)*. *lin-66(lf); acr-2(gf)* double mutants showed convulsions indistinguishable from *acr-2(gf)*. In *eif-3.G(gf); acr-2(gf)* background, *lin-66(lf)* restored convulsions to the level similar to *acr-2(gf)* single mutants (Fig 1B). We observed similar suppression effect in L4 animals of *eif-3.G(gf); lin-66(0); acr-2(gf)*. However, adult escapers of *eif-3.G(gf); lin-66(0); acr-2(gf)* showed less suppression effects, likely reflecting overall unhealthiness of *lin-66(0)*. We transgenically expressed WT *lin-66* genomic DNA, containing 2 kb upstream sequences, the full coding region, and 657-nt 3′ untranslated sequences, and observed complete reversal of the suppression effect associated with *lin-66(lf)* (Fig 1B). Together, these data indicate that loss of function in *lin-66* specifically reduces the gain-of-function effect of *eif-3.G(gf)* on *acr-2(gf)* convulsion behavior.

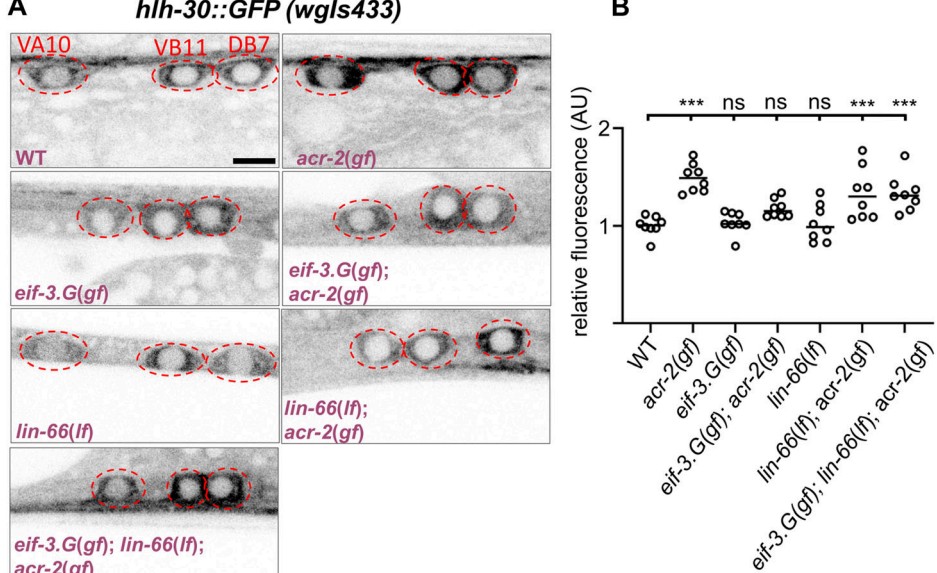

**A**  *hlh-30::GFP (wgIs433)*

VA10  VB11  DB7

WT   acr-2(gf)

eif-3.G(gf)   eif-3.G(gf); acr-2(gf)

lin-66(lf)   lin-66(lf); acr-2(gf)

eif-3.G(gf); lin-66(lf); acr-2(gf)

**B**

**Figure 2.** *lin-66* modulates protein translation in cholinergic motor neurons, dependent on *eif-3.g(gf)* and *acr-2(gf)*.
**(A)** Representative confocal images of the cholinergic motor neuron somata from animals expressing *hlh-30::GFP* (a fosmid-based reporter, *wgIs432*). Cholinergic motor neurons were identified based on their stereotypic position, aided by a *Pacr-2-mCherry* reporter; the somata were enclosed by a dotted red line; scale = 4 μm. Alleles used are the same as in Fig 1A. **(B)** Quantification of HLH-30::GFP intensity from each genotype (n = 8). Each dot represents the average fluorescence intensity from the VA10, VB11, and DB7 neuronal somata normalized to the WT control. Statistics: *** *P* < 0.001 and ns, not significant, one-way ANOVA with Bonferroni's post hoc test.

## LIN-66 acts in cholinergic motor neurons to mediate EIF-3.G(gf) activity on protein translation

*eif-3.G* and *lin-66* are broadly expressed in many types of cells (Morita & Han, 2006; Blazie et al, 2021). We previously showed that the suppression of *acr-2(gf)* convulsion by *eif-3.g(gf)* is due to a cell-autonomous activity of *eif-3.g(gf)* in the cholinergic motor neurons (Blazie et al, 2021). To test the cell-type requirement of *lin-66*, we expressed full-length cDNAs of *lin-66 (isoform lin-66.A or lin-66.C)* specifically in cholinergic motor neurons using the *Punc-17β* promoter because these two mRNA isoforms include exon 4 where the *ju1661* mutation is located. We found that *eif-3.G(gf); lin-66(lf); acr-2(gf)* animals carrying such transgenes exhibited WT locomotion resembling *eif-3.G(gf); acr-2(gf)* (Fig 1C). These data indicate that *lin-66* acts in the cholinergic motor neurons to interfere with the *eif-3.G(gf)* function.

We previously reported that EIF-3.G(gf) selectively modulates protein translation efficiency on mRNAs with GC-rich 5′ UTRs to suppress convulsions of *acr-2(gf)* (Blazie et al, 2021). We next asked whether the genetic interaction between *lin-66* and *eif-3.G(gf)* involves regulation of protein translation by examining the expression of an EIF-3.G target, HLH-30. In *acr-2(gf)* single mutants, the fluorescence intensity of HLH-30::GFP from a fosmid reporter (*wgIs433*) in the ventral cord cholinergic motor neurons was selectively and significantly elevated, and in *eif-3.G(gf); acr-2(gf)* double mutants, HLH-30::GFP expression in these neurons was reduced to levels in WT animals (Fig 2) (Blazie et al, 2021). We found that in *lin-66(lf)* single mutants, HLH-30::GFP intensity in the cholinergic motor neurons was similar to that in WT animals. However, in *eif-3.G(gf); lin-66(lf); acr-2(gf)* animals, HLH-30::GFP expression was increased to the level comparable to that in *acr-2(gf)* single mutants (Fig 2). In addition, *lin-66(lf)* did not further increase HLH-30::GFP expression in *acr-2(gf)* single mutants (Fig 2). Therefore, these data support the conclusion that *lin-66* mediates the activity of *eif-3.G(gf)* on protein translation.

## Expression of LIN-66 and EIF-3.G is largely independent of each other

We considered two possible explanations for how loss of *lin-66* function may reduce the activity of *eif-3.G(gf)*. One is that *eif-3.G(gf)* might cause an increased expression of LIN-66; alternatively, *eif-3.G(gf)* expression is dependent on *lin-66*. We first examined a previously published transgenic overexpression reporter line of *lin-66* (Morita and Han, 2006), and observed a comparable expression pattern in WT and *eif-3.G(gf)*. To more precisely determine LIN-66 expression, we used CRISPR editing technology to tag the endogenously expressed LIN-66 with GFP fused in-frame at the C-terminus, allele designated *ju2002* (see the Materials and Methods section). However, the fluorescence intensity of the knock-in (KI) LIN-66::GFP was nearly invisible, indicating that LIN-66 is not abundantly expressed. We noticed that an intermediate GFP knock-in product that retained the *let-858* 3′ UTR used in the knock-in vector (Fig 3A, allele designated *ju1934*) showed detectable fluorescence in muscle, epidermis, and neuronal tissues throughout all developmental stages. Despite the increased expression, *lin-66::GFP(ju1934)* behaved as WT, as homozygous *lin-66::GFP(ju1934)* animals were healthy and did not show any suppression effects on *eif-3.G(gf); acr-2(gf)*. In the ventral cord motor neurons, LIN-66::GFP fluorescence was mostly concentrated within the somatic cytoplasm. The overall intensity and subcellular pattern of *lin-66(ju1934)* were not altered in the presence of *eif-3.G(gf)*, or in *eif-3.G(gf); acr-2(gf)* double mutants (Fig 3B). These observations argue against the idea that EIF-3.G(gf) regulates translation of LIN-66. Moreover, *lin-66(ju1934); acr-2(gf)* animals showed convulsion similar to *acr-2(gf)*, suggesting that simply increasing LIN-66 expression is not sufficient to mimic the activity of EIF-3.G(gf). Conversely, we tested whether *lin-66* might regulate the expression of EIF-3.G. We found that functional GFP::EIF-3.G expressed from an integrated single-copy transgene showed similar intensity and localization pattern in WT and *lin-66(ju1661)*.

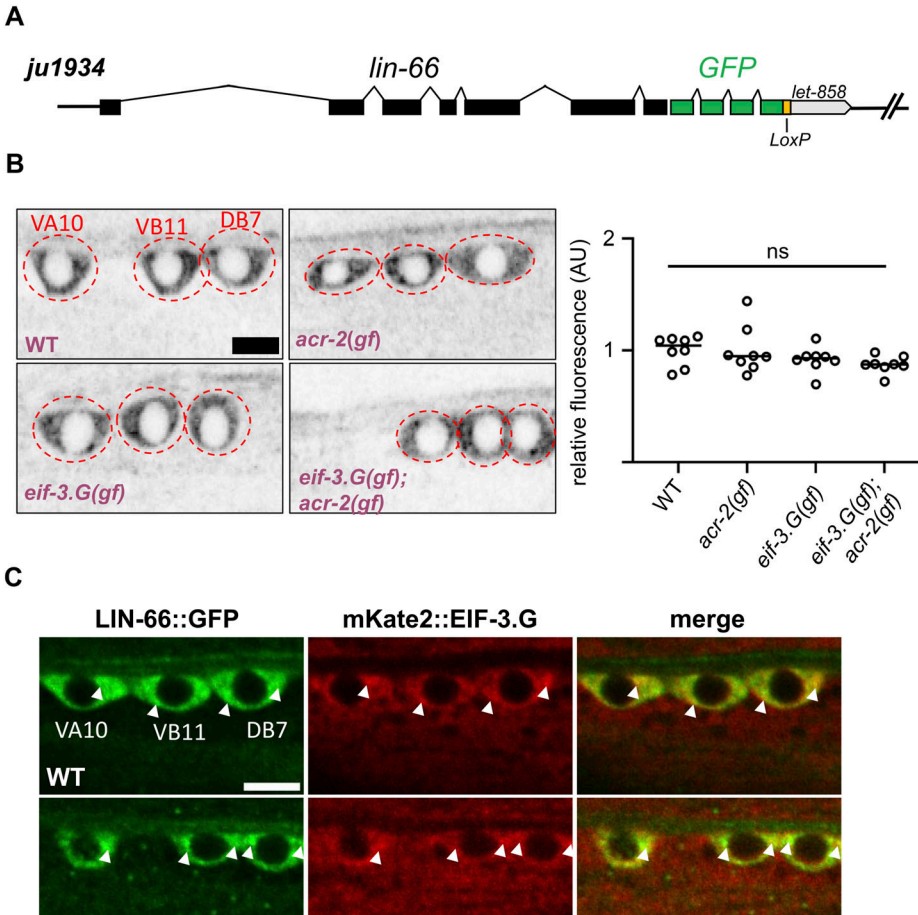

**Figure 3. LIN-66 localizes predominantly to the somatic cytoplasm, and its expression pattern is not affected by *eif-3.g(gf)* or *acr-2(gf)*.**
**(A)** *ju1934* has GFP knock-in to the C-terminus of *lin-66*, and retains the *let-858* 3′ UTR followed by the self-excision cassette flanked by LoxP sites. **(B)** Representative confocal images of LIN-66::GFP*(ju1934)* in the indicated cholinergic motor neuron soma (left) and quantification of GFP intensity in the three neuronal somata in each genotype (right, n = 8 animals, scale = 4 *µm*). Statistics: one-way ANOVA with Bonferroni's post hoc test, ns, not significant. Alleles used are the same as in Fig 1. **(C)** Representative confocal images of LIN-66::GFP and mKate2::EIF-3.G expression in indicated cholinergic motor neuron somata; white arrows point to their co-localization in WT and in *acr-2(gf)*; scale = 4 *µm*.

To further address whether the subcellular localization pattern of EIF-3.G and LIN-66 might depend on each other, we co-expressed an integrated single-copy transgene of *Peif-3.G-mKate2::eif-3.G(WT)* with *lin-66::GFP(ju1934)*. By confocal imaging analysis of the ventral cord motor neurons, we observed that the two proteins showed a largely overlapping granular pattern and that the co-localization pattern of LIN-66::GFP and mKate2::EIF-3.G remained similar in both WT and *acr-2(gf)* animals (Fig 3C). Based on these data, we conclude that LIN-66 and EIF-3.G unlikely regulate each other's expression in the ventral cord motor neurons.

### LIN-66 localization and function depend on both a predicted structural region and low-complexity sequences

LIN-66 was previously reported to exhibit homology across nematode clades but with no apparent mammalian orthologs or identifiable domains. We therefore employed in silico tools to further analyze the LIN-66 protein sequence (see the Materials and Methods section). SMART protein domain analysis revealed several low-complexity amino acid sequences at the N- and C-terminal region of LIN-66 (Fig S1), which was also predicted to be highly intrinsically disordered by Predictors of Natural Disordered Regions. Additional prediction of folded protein structures using AlphaFold suggested amino acids 101–370 in the middle of LIN-66 to be a structured region (Figs 4A and S1).

To test the functional relevance of the prediction, we generated a series of LIN-66 translation reporters using *lin-66.C* cDNA fused to GFP at the C-terminus and driven by the *Punc-17β* promoter, which is active only in the cholinergic motor neurons (Fig 4A). Overexpressed full-length LIN-66::GFP showed somatic cytoplasmic localization with a punctate pattern (Fig 4B). Removing the N-terminal disordered sequences, LIN-66(Δ10–44), did not change the overall localization pattern. Removing low-complexity sequences at the C-terminus, LIN-66(Δ421–554), caused the protein to appear more diffuse (Fig 4B). Removing both N- and C-terminal low-complexity sequences, LIN-66(Δ10–44 + Δ421–554), showed slightly increased expression but remained diffuse like LIN-66(Δ421–554) (Fig 4B). In *eif-3.g(gf); lin-66(lf); acr-2(gf)* triple mutant animals, all three LIN-66 mutant constructs showed rescuing activity, similar to the full-length LIN-66 transgene (Fig 4C). In contrast, a transgene containing only the predicted structured region, LIN-66(Δ2–90, Δ387–624), showed a diffuse localization pattern and did not exhibit rescuing activity (Fig 4B and C). Other LIN-66::GFP transgenes that omitted amino acids either in the N-terminus, LIN-66(Δ10–172), or in the C-terminus, LIN-66(Δ298–367), or nearly the entire predicted structural region, LIN-66(Δ116–367), did not affect the overall

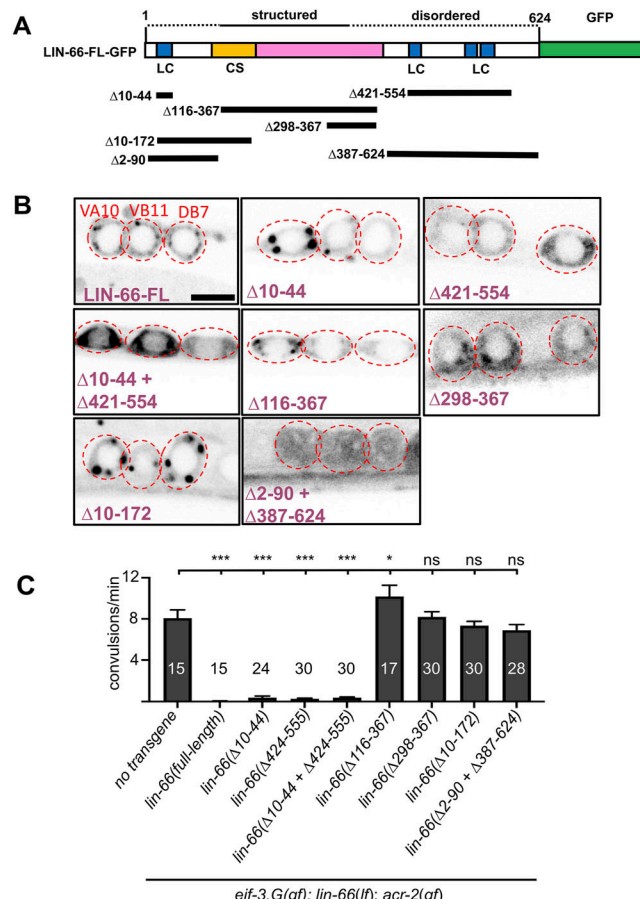

**Figure 4. LIN-66 function depends on both the predicted structured region and flanking low-complexity and intrinsic disordered sequences.**
**(A)** Illustration of the LIN-66 protein (isoform C) tagged with GFP at the C-terminus. The colored domains include LC for the low-complexity sequence and CS for the cold-shock domain. Dotted lines above represent predicted disordered regions, and the solid line, the structured region predicted by AlphaFold. Black bars below depict sequences deleted in *lin-66* truncation transgenes. **(B)** Representative images of the indicated cholinergic motor neuron somata expressing each *lin-66*-GFP transgene illustrated in (A): FL, full length; Δ indicates the amino acids deleted; scale = 4 μm. **(C)** Quantification of convulsion in animals expressing the *lin-66* transgenes shown in panels (A, B). At least two transgenic lines for each expression construct were analyzed Statistics: one-way ANOVA with Bonferroni's post hoc test, ***$P < 0.01$, *$P < 0.05$, and ns, not significant.

localization pattern and also had no rescue activity (Fig 4B and C). These data indicate that the regions outside of the predicted structured region can influence LIN-66 punctate pattern and that the predicted structured region is necessary, but insufficient, for LIN-66 function in *eif-3.g(gf)*-mediated behavioral suppression of *acr-2(gf)*.

### LIN-66 function depends on a cold-shock domain

To further identify functional homologous domains of the structured region in LIN-66, we inputted the AlphaFold LIN-66 structure to the ProFunc server, which predicts protein function from structure (Laskowski et al, 2005). This analysis revealed that amino acids 101–170 are predicted to fold into a cold-shock domain

(Fig 5A), most similar to those present in *C. elegans* LIN-28 and human LIN-28 and CSDE1 (Figs 5B and S2) (Jacquemin-Sablon et al, 1994; Moss et al, 1997). The observation that LIN-66(Δ10–172)::GFP did not show rescuing activity in *eif-3.g(gf), lin-66(lf); acr-2(gf)* (Fig 4B and C) supported the functional importance of the predicted cold-shock domain.

Cold-shock domains generally consist of five β-strands, with the second and third β-strands containing highly conserved amino acids necessary for RNA binding (Heinemann & Roske, 2021). By amino acid sequence alignment of LIN-66 isoform C, the predicted cold-shock domain of LIN-66 showed significant homology in the conserved regions, notably the conserved glycine in the β1-strand and the conserved amino acid residues within the RNP1 and RNP2 motifs (Fig 5C). To test whether the residues in RNP motifs are important for LIN-66 function, we mutated them in the full-length cDNA LIN-66::GFP translation reporter, designated *lin-66(CSD*)*. When expressed in the cholinergic motor neurons, we observed that LIN-66(CSD*)::GFP exhibited a diffuse pattern with prominent puncta in neuronal somata (Fig 5D), similar to that observed with full-length LIN-66. However, the *lin-66(CSD*)* transgene did not show any rescuing activity in *eif-3.G(gf); lin-66(lf); acr-2(gf)* triple mutants (Fig 5E). Taken together, these data show that the predicted cold-shock domain in LIN-66 is required for its function in EIF-3.g(gf)–dependent regulation of neuronal hyperexcitation, likely involving its RNA-binding activity.

## Discussion

The eIF3 complex possesses a variety of functions to modulate protein synthesis by engaging the unique activities of its 13 subunits (Cate, 2017). We previously showed that *C. elegans* EIF-3.G/eIF3g conveys a specific regulation of protein translation via the conserved zinc finger to modulate neuronal activity of the cholinergic motor neurons (Blazie et al, 2021). We now provide evidence that this specialized activity of EIF-3.G/eIF3g requires LIN-66, a protein previously described to regulate protein translation through an unknown mechanism during animal development (Morita & Han, 2006). Furthermore, by combining structural informatics analysis with in vivo functional dissection, we have uncovered a cold-shock domain in LIN-66 that is critical for its function. We propose that LIN-66 acts as a cell type– and context-dependent facilitator to mediate the activity of EIF-3.G/eIF3g in protein translation.

*C. elegans lin-66* was discovered in genetic screening for genes regulating the vulva cell fate. Null mutants of *lin-66* are arrested in late-stage larvae, indicating that it has essential roles in animal development (Morita & Han, 2006). *lin-66* exhibits complex genetic interactions with multiple genes known to regulate temporal cell fate specification. Moreover, in *lin-66* null mutants the expression of the cold-shock domain protein LIN-28 is increased, which may partly involve the 3′ UTR of *lin-28* mRNA. The authors proposed that *lin-66* may act in a post-transcriptional regulatory network to modulate protein translation, although they speculated that *lin-66* might not directly act on *lin-28* mRNA. In our *acr-2(gf)*–induced cholinergic overexcitation paradigm, through unbiased genetic screening, we were able to isolate a partial loss-of-function

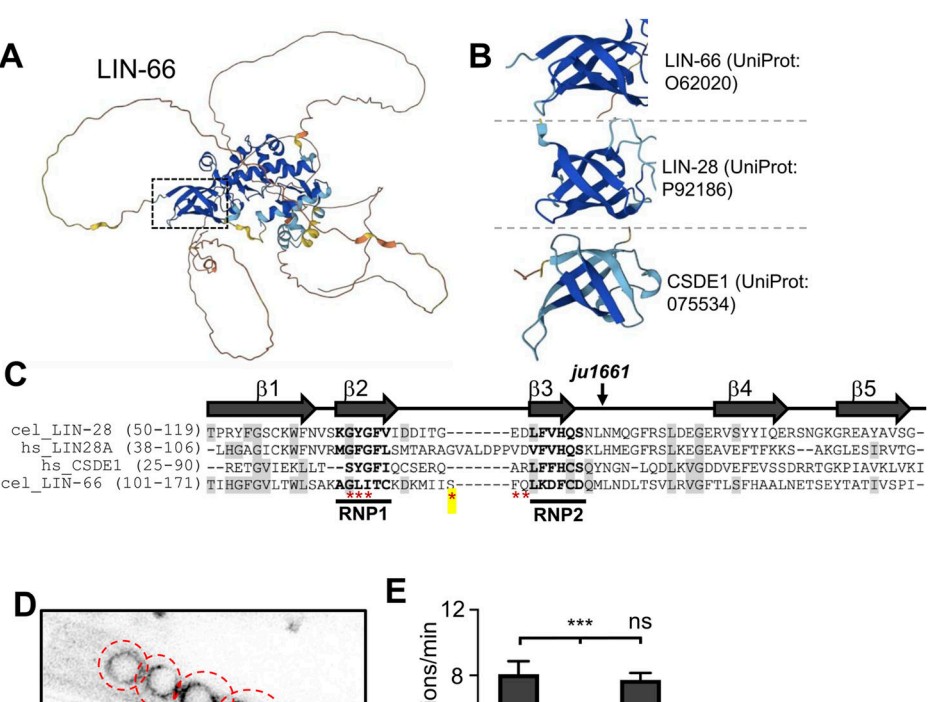

**Figure 5. Cold-shock domain in LIN-66 is important for its function.**
**(A)** Predicted structure of LIN-66 protein (UniProt accession: O62020) generated by the AlphaFold algorithm (https://alphafold.ebi.ac.uk/), with the box pointing to the cold-shock domain predicted by ProFunc analysis (Laskowski et al, 2005). **(B)** Predicted LIN-66 cold-shock domain resembles that in *C. elegans* LIN-28 and the first cold-shock domain of human CSDE1. **(C)** Amino acid sequence alignment of cold-shock domains of *C. elegans* LIN-66 and LIN-28, and human LIN-28 and CSDE1. Bolded residues in β-strands 2 and 3 indicate the RNP1 (R/K)-G-(F/Y)-(G/A)-(F/Y)-V-X-(F/Y) and RNP2 (L/I)-(F/Y)-(V/I)-X-(N/G)-L consensus sequences. Red asterisks represent the amino acids mutated to alanine or serine in the *lin-66(CSD\*)* transgene. *ju1661* generates transcripts that would lead to LIN-66 out-of-frame beginning at the indicated amino acid within the CSD. **(D)** Representative confocal images of the indicated cholinergic motor neuron soma (red dotted outline) expressing WT *lin-66(full-length)::GFP* or mutant *lin-66(CSD\*)::GFP* transgene; scale = 4 μm. **(E)** Quantification of convulsion from animals with indicated genotype and transgenes. At least two transgenic lines were scored. Statistics: one-way ANOVA with Bonferroni's post hoc test, \*\*\*P < 0.01 and ns, not significant.

mutation of *lin-66* that displays no discernable developmental defects, but specifically ameliorates the gain-of-function effect of EIF-3.G/eIF3g on locomotion behavior and on protein translation in cholinergic motor neurons. Thus far, we have not yet found any evidence supporting a role of *lin-28* in *acr-2(gf)*–induced neuronal overexcitation. Our transgenic expression studies support the conclusion that the genetic interaction between *lin-66* and *eif-3.G* is specific to the cholinergic neuron hyperexcitation. We also show that LIN-66 mediates protein translation modulation through EIF-3.G targets that have GC-rich 5′ UTRs. We previously showed that elevated translation of HLH-30 in the cholinergic motor neurons depends on *eif-3.G* (Blazie et al, 2021). Here, we extend this observation with evidence that *eif-3.G(gf)*-mediated dampening of HLH-30 expression in the same cells depends on *lin-66*. It is worth noting that HLH-30::GFP images showed some variability in fluorescence intensity and punctate pattern among the VA10, VB11, and DB7 neuronal somata, although the mean expression for all three neurons was significantly different between genetic backgrounds. Thus, we consider that this neuron-specific variation is unlikely to explain expression differences because of genetic alterations. LIN-66 and EIF-3.G also co-localize to the same subcellular domains in motor neuron somata, and do not appear to regulate each other's

expression. We thus hypothesize that the two proteins provide functional co-dependence for regulation of protein translation.

Little is known about the function of LIN-66, which is likely due to both the lack of identifiable protein domains and the low abundance of LIN-66 protein (Morita & Han [2006] and this study). Through in silico analyses of available structural and informatics databases, we found that LIN-66 is predicted to have a structured region embedded within low-complexity sequences. Within the predicted structured region, we further identified a cold-shock domain. Cold-shock domains were originally named because of their presence in multiple bacterial proteins that were induced by cold shock (Sommerville, 1999; Mihailovich et al, 2010; Lindquist & Mertens, 2018; Heinemann & Roske, 2021). It is now known that cold-shock domains are present in many proteins with diverse functions, independent of any response to cold shock. For example, human Y-box binding protein 1 participates in transcription, splicing, and translation (Kleene, 2018). The LIN-28 proteins use a combination of their cold-shock domain with CCHC zinc fingers to regulate microRNA metabolism or processing (Moss & Tang, 2003). Other cold-shock domain–containing proteins can act as RNA chaperones or facilitate RNA processing or degradation (Lindquist & Mertens, 2018). Interestingly, several protein translation factors, such as

eIF1A, eIF2α, or eIF5A, all contain a cold-shock domain (Amir et al, 2018), although the function of these cold-shock domains remains undefined. Our in vivo structure–function dissection studies support the predicted LIN-66 protein structure. We find that mutating the conserved residues in the RNP motifs within the cold-shock domain abrogates LIN-66 function, without altering the overall protein expression, suggesting that LIN-66 has the potential to directly bind nucleic acids. Full-length LIN-66 localizes to the somatic cytoplasm with visibly detectable puncta or granules. We find that the low-complexity sequences of LIN-66 contribute to its punctate pattern. LIN-66 function depends on both the structured region and some low-complexity sequences. Low-complexity sequences are generally intrinsically disordered, and many studies have shown that proteins with intrinsically disordered regions, most notably those involved in forming RNA granules, have the propensity to undergo protein phase separation (Lee et al, 2022). Conceivably, LIN-66 function may engage in multivalent types of interactions to modulate protein synthesis by co-opting translation machinery in specialized cellular contexts. We often observed that punctum size and abundance positively correlated with the LIN-66 expression level, consistent with the concentration-dependent behavior of biophysical phase separation (Shin & Brangwynne, 2017). This observation was most evident when LIN-66 was expressed from extrachromosomal transgenic arrays, which are known to lead to mosaic expression. When expressed from these transgenes, LIN-66 puncta were generally more visible in cells where it was highly expressed and appeared diffusely localized in cells with lower expression (Fig 4B).

The precise function of LIN-66 in the context of cholinergic motor neurons remains to be determined. Our finding that LIN-66 cooperates with the translation initiation factor EIF-3.G and harbors a cold-shock domain is consistent with a role in regulating gene expression. Seven other C. elegans proteins that contain a cold-shock domain have been reported to have tissue-specific functions. LIN-28 is extensively studied and regulates both RNA stability and translation, depending on the cell-type and cellular context (Balzer & Moss, 2007; Heo et al, 2008). The Y-box proteins CEY-1 through CEY-4 regulate RNA stability in germ line (Arnold et al, 2014), and DIS-3 and DISL-2 appear to interact with microRNA and participate in the development of multiple tissues (Ustianenko et al, 2013; Weaver et al, 2014; Szczepińska et al, 2015). These studies highlight that mRNA turnover often follows tissue-specific mechanisms. Based on our data and the study by Morita & Han [2006], we speculate the mechanisms involving LIN-66 may depend on a cell-type and stimulus context. LIN-66 may regulate mRNA stability in neurons, adjusting the dosage of mRNA templates available for EIF-3.G–mediated translation. This model is supported by our finding that LIN-66 modulates EIF-3.G–dependent expression levels of HLH-30 (Fig 2). Alternatively, as LIN-66 and EIF-3.G co-occupy the same cellular space in cholinergic motor neurons (Fig 3C), LIN-66 may mediate protein translation, possibly via a direct interaction with EIF3.G. Identifying additional interacting partners of LIN-66 and EIF3.G may shed light on these models.

Dysregulation of protein translation is associated with many human neurological disorders (Kapur et al, 2017). The altered expression of eIF3g is linked to narcolepsy and autism (Holm et al, 2015; Choi & An, 2021). Cold-shock domain–containing proteins are also broadly associated with RNA metabolism and protein translation

(Lindquist & Mertens, 2018). Through unbiased forward genetic screening for locomotion behaviors associated with altered neuronal circuit activity, we have uncovered a highly selective protein translation regulatory network involving the interaction between the conserved EIF-3.G subunit and a cold-shock domain–containing protein LIN-66. Our findings underscore the power of forward genetic analysis in deciphering functional specificity underlying how an essential component of the protein translation complex can tune the neuronal proteome to balance neural circuit outputs.

# Materials and Methods

## C. elegans genetics

All C. elegans strains were maintained at 20°C on nematode growth media (NGM) seeded with E. coil OP50 as described previously (Brenner, 1974). Behavioral and imaging analyses were performed in young adult or L4 hermaphrodites; males were used for crosses. Compound mutants were generated by genetic crossing using standard procedures. Mutant alleles were detected using a combination of visual observations of locomotion and/or fluorescence reporters, and further verified by identification of allele-specific DNA changes using PCR and sequencing. All strains used in this study are listed in Table S1. Information on mutant alleles and the primers for PCR amplification is listed in Table S2.

### Genetic screening for suppressors of eif-3.G(gf)

We conducted a clonal genetic suppressor screening using CZ21759 eif-3.G(ju807gf); acr-2(n2420gf) double mutants, following the standard procedure (Brenner, 1974). Briefly, mix-staged eif-3.G(gf); acr-2(gf) double mutants were washed into a 2 ml solution of M9 media with 47 mM ethyl methanesulfonate (Sigma-Aldrich) and incubated at 22°C on a spinning wheel for 4 h. Worms were pelleted in a Beckman Allegra X-14R centrifuge at 523g for 2 min and washed twice with M9 media, with centrifugations in between washes. Worms were transferred to a fresh seeded NGM plate and allowed to recover for 20 min at 22°C. Mutagenized L4 (P0) animals were picked to 17 plates (2P0/plate) and cultured at 20°C. After 3 d, 18 F1 animals from each P0 plate (total 306 F1s) were singly transferred to freshly seeded NGM plates, and cultured at 20°C for two more days. Young adult F2 animals on each F1 plate were screened for any restored convulsion behavior by visual inspection under a dissection microscope. Many initial isolates did not produce viable progeny, and seven F2 animals produced true-breeding lines, but with variable convulsions resembling acr-2(gf) single mutants. We outcrossed the original isolates using N2 males and were able to re-isolate one suppressor, ju1661, that showed consistent suppression and that was dependent on both eif-3.g(gf) and acr-2(gf).

### Mapping of lin-66(ju1661)

Genomic DNAs were purified from strains CZ26710 and CZ26711 using Gentra Puregene Tissue Kit (QIAGEN). Whole-genome DNA library preparation and 90-bp paired-end sequencing were performed by

Beijing Genomics Institute (BGI). Sequencing reads of 20X coverage were mapped to the *C. elegans* reference genome (ce10) using Burrows–Wheeler Aligner (Li & Durbin, 2009). Genome Analysis Toolkit (McKenna et al, 2010) was used to locate SNPs in the mapped reads compared with the reference genome. The mapped SNPs were annotated by their relative proximity to the nearest gene model (WS220) using SnpEff version 2.1a (Cingolani et al, 2012). We used custom SQL scripts to identify SNPs that are detected in both the original isolate (CZ26710) and the 1x outcross isolate (CZ26711) and excluded from CZ21759 (sequences reported in Blazie et al [2021]). We performed further outcrossing to N2 and re-isolated multiple *ju1661* recombinants. Using the mutagenesis-induced SNPs to perform fine mapping of these recombinants, we located *ju1661* to a small interval of chromosome IV containing *lin-66*.

### CRISPR editing to knock in GFP to *lin-66*

We used the CRISPR/Cas9 SEC (self-excision cassette) method (Dickinson et al, 2013) to insert the GFP in-frame immediately preceding the stop codon in the endogenous *lin-66* locus. Homology arm sequences that included 500 bp upstream (primers YJ12796 and YJ12797) and 500 bp downstream (primers YJ12794 and YJ12795) of the *lin-66* stop codon were PCR-amplified from N2 genomic DNA. Silent mutations were introduced in the portion of the homology arm containing the sgRNA sequence to prevent Cas9 cleavage of the repair template. The Gibson assembly (NEB) was performed to ligate the resulting PCR amplicons with the pDD268 vector (#132523; Addgene) digested with AvrII and SpeI, producing the clone pCZ1053. The sgRNA sequence 5′-CGTTTGAACTCACTCCGTAT-3′ was incorporated into the pDD162 vector to produce pCZ1054, following the protocol as described previously. A DNA mix containing 10 ng/$\mu$l pCZ1053, 50 ng/$\mu$l pCZ1054, 10 ng/$\mu$l pGH8, 5 ng/$\mu$l pCFJ104, and 2.5 ng/$\mu$l pCFJ90 was injected into N2 young adult hermaphrodites. The P0 animals were cultured at 25°C for 3 d, and their progeny were treated with 250 $\mu$g/$\mu$l hygromycin. After another 3 d, surviving progeny that showed the roller phenotype but did not express co-injection mCherry markers were selected. A single line, designated *ju1934*, was confirmed by PCR genotyping to contain GFP in-frame to the last amino acid of LIN-66. To remove SEC, ~16 CZ29347 *lin-66(ju1934)* L1 animals were transferred to a freshly seeded NGM plate and heat-shocked at 34°C for 6 h. After culture at 22°C for 3 d, animals without the rolling phenotype were selected and verified for the presence of GFP and absence of the SEC, resulting in an allele designated *ju2002*.

### Molecular cloning and transgenic expression of *lin-66* WT and variants

Full-length *lin-66* genomic DNA including 2 kb upstream sequences and 650 nt downstream of the stop codon was obtained by PCR amplification from WT (N2) genomic DNA and cloned into the Gateway entry vector to generate pCZGY3547. cDNAs for *lin-66A* and *lin-66C* were obtained from WT cDNA library and cloned into the Gateway entry vector to generate pCZGY3561 and pCZGY3560, respectively. GFP fusion expression constructs were made by the Gibson assembly using primers listed in Table S3. All truncated (Δ) *Plin-66::lin-66C::GFP* expression constructs were generated using the phosphorylated primers listed in Table S3. Briefly, primer pairs

for each Δ DNA fragment were used to PCR-amplify a portion of pCZGY3555 (*Plin-66-lin-66C::GFP*) and the product was treated with DpnI to remove the template and DpnI deactivated at 80°C for 25 min. The resulting product was then subjected to intramolecular ligation using T4 DNA ligase and transformed into DH5α. All final clones were verified by Sanger sequencing and listed in Table S3. A mixture of 20 ng/$\mu$l transgene and 70 ng/$\mu$l 1 kb ladder (NEB) filler DNA was microinjected into CZ26711 young adult hermaphrodites, following the standard procedure (Mello et al, 1991). More than two transgenic lines were selected by screening F1 progeny of micro-injected animals for GFP expression and transmission efficiency.

### In silico annotation of LIN-66 protein structure

We used the simple modular architecture tool (http://smart.embl-heidelberg.de/ [Letunic et al, 2015]) and SEG (https://mendel.imp.ac.at/METHODS/seg.server.html [Wootton, 1994]) to locate low-complexity (LC) amino acid sequences in LIN-66 isoform C. Predictors of Natural Disordered Regions (http://www.pondr.com/ [Xue et al, 2010]) was used with default settings to assess disordered sequence regions of LIN-66. The structured region between amino acids 101–376 and cold-shock domains (amino acids 101–171) was identified from the output of AlphaFold software (https://alphafold.ebi.ac.uk/) using LIN-66 (UniProt: A0A061AL58) with default settings (Jumper et al, 2021). Structural homology between the cold-shock domain of LIN-66 and human CSDE1 (Fig S2) was identified by inputting the entire LIN-66 isoform C amino acid sequence into the ProFunc database (http://www.ebi.ac.uk/thornton-srv/databases/profunc [Laskowski et al, 2005]) using default settings.

### Quantification of convulsion behavior

Convulsion behavior was assessed by visual inspection of day 1 young adult hermaphrodites under a dissection microscope according to the following protocol. L4 animals were transferred to a freshly seeded NGM plate the day before behavioral assay. The next day, each single young adult animal was placed onto the bacterial lawn of an NGM plate seeded with OP50 and allowed to acclimate for 90 s. Convulsions, defined as a brief shortening of the animal body length accentuated by muscle contraction, were quantified over a 90-s period of continuous monitoring. Observers were blinded to genotype. Convulsions in *lin-66(0)* animals were quantified at the L4 stage because of their developmental arrests before reaching adult. Our data report the average convulsion frequency over a 60-s interval.

### Confocal imaging and analysis

L4 or young adult animals were immobilized in a small droplet of 1 mM levamisole in M9 liquid media on a microscope slide containing a 2% agarose pad. Imaging was performed with a Zeiss LSM800 confocal microscope using 63X lens and identical image settings (1.25 mm pixel size with 0.76-ms pixel time, 50-mm pinhole), unless otherwise indicated. We used *juEx2045(Pacr-2-mCherry)* to identify the VA10, VB11, and DB7 cholinergic motor neurons as previously described (Blazie et al, 2021). All other imaged cell types were identified based on their stereotyped position and anatomical features. Observers were blinded to the genotype when possible.

## Quantifying fluorescence in motor neuron somata

All quantification of fluorescence intensity was performed using the integrated density function in ImageJ (Schindelin et al, 2012). We acquired the mean integrated density from an ROI focused around each of the VA10, VB11, and DB7 somata and subtracted the background from a region of each image lying outside of the specimen. The integrated density values obtained from each cell were then averaged to the value expressed in each data point of Figs 2B and 3B. The resulting values were normalized to the mean fluorescence intensities obtained from the same GFP reporter in the same neuronal soma in a WT background.

## Statistical analysis

GraphPad Prism 9 software was used for all statistical analyses, and $P$-values below 0.05 were considered significant. Sample sizes were determined via power analysis as in our previous studies (Takayanagi-Kiya et al, 2016; McCulloch et al, 2017; Blazie et al, 2021).

# Data Availability

Whole-genome sequence reads and fluorescence intensity measurement data are available upon request.

# Supplementary Information

# Acknowledgements

We thank our laboratory members for valuable discussions throughout the work and M Han for providing some of the *lin-66* reagents. Some strains were provided by the CGC, which is funded by NIH Office of Research Infrastructure Programs (P40 OD010440). SM Blazie and D Fortunati were trainees on the NINDS T32 training grant (NS007220). This work was supported by R37 NS035546 and R35 NS127314 (to Y Jin).

## Author Contributions

SM Blazie: conceptualization, data curation, formal analysis, investigation, methodology, and writing—original draft, review, and editing.
D Fortunati: investigation and writing—review and editing.
Y Zhao: data curation and investigation.
Y Jin: conceptualization, resources, supervision, funding acquisition, and writing—original draft, review, and editing.

## Conflict of Interest Statement

The authors declare that they have no conflict of interest.

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
