## [Reviewer comments · Life Science Alliance]

Life Science Alliance

C. elegans LIN-66 mediates EIF-3 G/eIF3-dependent protein translation via a cold-shock domain

Stephen Blazie, Daniel Fortunati, Yan Zhao, and Yishi Jin

DOI: <https://doi.org/10.26508/lsa.202402673>

Corresponding author(s): Yishi Jin, University of California, San Diego

Review Timeline:

Submission Date:	2024-02-23
Editorial Decision:	2024-05-01
Revision Received:	2024-05-26
Editorial Decision:	2024-05-29
Revision Received:	2024-05-30
Accepted:	2024-06-04

Transaction Report:

May 1, 2024

Re: Life Science Alliance manuscript #LSA-2024-02673-T

Prof. Yishi Jin
University of California, San Diego
Division of Biological Sciences University of California, San Diego Howard Hughes Medical Institute
La Jolla, CA 92093

Dear Dr. Jin,

Thank you for submitting your manuscript entitled "C. elegans LIN-66 mediates EIF-3.G-dependent protein translation via a cold-shock domain" to Life Science Alliance. The manuscript was assessed by expert reviewers, whose comments are appended to this letter. We invite you to submit a revised manuscript addressing the Reviewer comments.

Thank you for this interesting contribution to Life Science Alliance. We are looking forward to receiving your revised manuscript.

Sincerely,

B. MANUSCRIPT ORGANIZATION AND FORMATTING:

Reviewer #1 (Comments to the Authors (Required)):

In this manuscript, Blazie and colleagues performed a genetic screen to search for factors that reverted the suppression of eif-3.G(gf) on the convulsion of *acr-2(gf)* animals. They identified a loss of function mutation in *lin-66*. Based on structure prediction, the authors identified a putative cold-shock domain in LIN-66 and presented evidence that this domain is necessary for protein function in specific mutants/neurons.

The manuscript presents a set of solid genetic experiments and interesting genetic interactions but falls short of clarifying LIN-66 function. The evidence that LIN-66 mediates EIF-3.g(gf)-dependent translation is based solely on HLH-30::GFP expression. I appreciate the difficulty of studying translation in single cells. Still, the proposition that the function is at the translation level remains highly speculative. I wonder: other *C. elegans* cold-shock domain proteins impact mRNA levels (Arnold et al., NAR, 2014). Could LIN-66 have a similar effect on *hlh-30* and other mRNAs?

Reviewer #2 (Comments to the Authors (Required)):

This manuscript reports the characterisation of a gain-of-function mutation in the G subunit of the eIF3 complex in *C. elegans*, known as *eif-3.g(gf)*, which selectively influences protein translation in ventral cord cholinergic motor neurons. Through unbiased genetic suppressor screening, they pinpointed the *lin-66* gene as a mediator of *eif-3.g(gf)*-dependent protein translation in these neurons. LIN-66 is a nematode-specific protein characterized by largely low-complexity amino acid sequences and previously unknown functional domains.

By combining bioinformatic analysis with *in vivo* functional dissection, researchers identified a cold-shock domain within LIN-66 that is crucial for its function. LIN-66 exhibits somatic cytoplasmic localization in cholinergic motor neurons, closely associating with EIF-3.G. The low complexity amino acid sequences of LIN-66 influence its subcellular distribution. Cold-shock domains, known for their interactions with RNA, play diverse roles in RNA metabolism and protein translation. The findings suggest that LIN-66 facilitates stimuli-dependent protein translation potentially by enhancing the interaction between mRNAs and EIF-3.G, shedding light on the intricate regulatory mechanisms underlying translation initiation in neuronal contexts.

The manuscript is well-presented and easy to follow.

Minor comments

Page 6. Why were *lin-66A* and *lin-66C* chosen for analysis?

Page 7/Figure 2B. The expression of the *hlh30::GFP* is quite variable between VA10, VB11, and DB7 neurons within individuals. No mention of this was made in the text, and the potential reasons for such variation were not provided. Some of the images showed small punta-how was this dealt with when determining relative fluorescence?

Page 8. It is easier for the reader if the genotypes of the mutants are written consistently between the text and the figures; *lin-66(lf)*; *eif-3.g(gf)*; *acr-2(gf)* figure *eif-3.G(gf)*; *lin-66(lf)*; *acr-2(gf)*

Page 8/Figure 4B: in the text 421-555, in the figure 421-554. Several deletions are not mentioned in the text. They should be removed if not discussed. In VA10 neurons, there appears to be more punta forming in several of the truncated proteins (10-44, 116-367 and 10-172) - this is not really addressed in the text.

Page 9. Which isoform of LIN-66 was analysed?

Point-to-point response

(italic shows Reviewer comments; Author's response shown in blue)

Reviewer #1:

The manuscript presents a set of solid genetic experiments and interesting genetic interactions but falls short of clarifying LIN-66 function. The evidence that LIN-66 mediates EIF-3.g(gf)-dependent translation is based solely on HLH-30::GFP expression. I appreciate the difficulty of studying translation in single cells. Still, the proposition that the function is at the translation level remains highly speculative. I wonder: other C. elegans cold-shock domain proteins impact mRNA levels (Arnold et al., NAR, 2014). Could LIN-66 have a similar effect on hlh-30 and other mRNAs?

We appreciate the reviewer for raising this point. We agree that our data lacks direct support for LIN-66 functions at the translation level and acknowledge the possibility that LIN-66 may also affect mRNA stability such as the germline functions of cey genes covered in (Arnold et al., 2014).

In the revised manuscript, we have expanded our discussion to incorporate the reviewer's concern. The new text is lines 372-391, marked in blue, and copied below:

"The precise function of LIN-66 in the context of cholinergic motor neurons remains to be determined. Our finding that LIN-66 cooperates with translation initiation factor EIF-3.G and harbors a cold shock domain is consistent with a role in regulating gene expression. Seven other C. elegans proteins that contain a cold-shock domain have been reported to have tissue-specific functions. LIN-28 is extensively studied and regulates both RNA stability and translation, depending on the cell type and cellular context (Balzer & Moss, 2007; Heo et al, 2008). The Y-box proteins CEY-1 through 4 regulate RNA stability in germ line (Arnold et al, 2014); and DIS-3 and DISL-2 appear to interact with microRNA and participate in the development of multiple tissues (Ustianenko et al, 2013; Weaver et al, 2014; Szczepińska et al, 2015). These studies highlight that mRNA turnover often follows tissue specific mechanisms. Based on our data and the study by Morita and Han (Morita & Han, 2006), we speculate the mechanisms involving LIN-66 may depend on a cell type and stimulus context. LIN-66 may regulate mRNA stability in neurons, adjusting the dosage of mRNA templates available for EIF-3.G mediated translation. This model is supported by our finding that LIN-66 modulates EIF-3.G dependent expression levels of HLH-30 (Figure 2). Alternatively, as LIN-66 and EIF-3.G co-occupy the same cellular space in cholinergic motor neurons (Figure 3C), LIN-66 may mediate protein translation, possibly via a direct interaction with EIF.3G. Identifying additional interacting partners of LIN-66 and EIF.3G may shed light to these models. "

Reviewer #2:

1) Page 6. Why were lin-66A and lin-66C chosen for analysis?

We chose *lin-66A* and *lin-66C* cDNAs for genetic dissection of LIN-66 function because they are the mRNA isoforms containing exon 4 where *ju1661* is located. The short isoform *lin-66B* consists of exons 6 and 7, unlikely sufficient for the function of *lin-66* related to *eif-3.G* activity in cholinergic motor neurons. This is also corroborated by our finding that removing amino acids 421-554 of LIN-66 isoform C, which covers most of *lin-66B* isoform, has no functional rescue activity (Figure 4B).

We have modified the text at lines 166-169 to clarify this point: "To test the cell type requirement of *lin-66*, we expressed full-length cDNAs of *lin-66.A* or *lin-66.C* specifically in cholinergic motor neurons using the *Punc-17b* promoter because these mRNA isoforms include exon 4 where the *ju1661* mutation is located."

2) Page 7/Figure 2B. The expression of the *hlh30::GFP* is quite variable between VA10, VB11, and DB7 neurons within individuals. No mention of this was made in the text, and the potential reasons for such variation were not provided. Some of the images showed small punta-how was this dealt with when determining relative fluorescence?

We thank the reviewer for raising this point. We did observe some variation in the expression levels and pattern between individual neurons VA10, VB11, and DB7. We note that this variability was observed in wild type (control) animals as well as *eif-3.G(gf)*, *lin-66(lf)*, and *acr-2(gf)* single mutants, i.e. the variability of HLH-30 expression is not specifically associated with a particular genotype. We averaged GFP intensity from individual neurons from an ROI drawn around each cell. The average GFP intensity values of the three cells were then averaged to the value presented in the dot plot in Figure 2B.

We now clarify our quantification protocol in the methods, lines 553-557, " We acquired the mean integrated density from an ROI focused around each VA10, VB11, and DB7 soma and subtracted background from a region of each image lying outside of the specimen. The integrated density values obtained from each cell were then averaged to the value expressed in each data point of Figures 2B and 3B."

We also added the following to our discussion lines 320-328, " We previously showed that elevated translation of HLH-30 in the cholinergic motor neurons depends on *eif-3.G* (Blazie et al., 2021). Here, we extend this observation with evidence that *eif-3.G(gf)*-mediated dampening of HLH-30 expression in the same cells depends on *lin-66*. It is worth noting that HLH-30::GFP images showed some variability in intensity and punctae localization among the VA10, VB11, and DB7 neuronal soma, although the mean expression for all three neurons was significantly different between genetic backgrounds. Thus, we consider that this neuron-specific variation is unlikely to explain expression differences due to genetic alterations."

3) Page 8. It is easier for the reader if the genotypes of the mutants are written consistently between the text and the figures; *lin-66(lf)*; *eif-3.g(gf)*; *acr-2(gf)* figure *eif-3.G(gf)*; *lin-66(lf)*; *acr-2(gf)*.

Thank you! We have made the necessary revisions to ensure consistent referencing of mutants by order of chromosome: *eif-3.g(gf)*; *lin-66(lf)*; *acr-2(gf)*.

4) Page 8/Figure 4B: in the text 421-555, in the figure 421-554. Several deletions are not mentioned in the text. They should be removed if not discussed. In VA10 neurons, there appears to be more puncta forming in several of the truncated proteins (10-44, 116-367 and 10-172) - this is not really addressed in the text.

Thank you for alerting us this error. The correct deletion span 421-554 is now reported in the manuscript text. We have also added the following text in lines 252-255 of the revised manuscript to dedicate mention to several deletions that were grouped together in the text of our first submission: " Other LIN-66::GFP transgenes that omitted amino acids either in the N-terminus, LIN-66(Δ 10-172), or in the C-terminus, LIN-66(Δ 298-367), or nearly the entire predicted structural region, LIN-66(Δ 116-367), did not affect the overall localization pattern and also had no rescue activity (Figure 4B-C)."

We agree with the reviewer that some of the truncated proteins appear to show more puncta in specific neuronal soma. We have generally observed that the visibility of *lin-66* punctae positively correlates with LIN-66 expression level. We expressed these *lin-66* transgenes used in our deletion study as extrachromosomal arrays, which convey mosaic expression that could lead to cell-specific variation in punctae formation.

We now address this observation in our discussion, lines 363-370: " We often observed that punctae size and abundance positively correlated with LIN-66 expression level, consistent with the concentration-dependent behavior of biophysical phase separation (Shin & Brangwynne, 2017). This observation was most evident when LIN-66 was mosaically expressed from extrachromosomal transgenic arrays, which are known to lead to mosaic expression. When expressed from these transgenes, LIN-66 punctae were generally more visible in cells where it was highly expressed and appeared diffusely localized in cells with lower expression (Figure 4B). "

6) Page 9. Which isoform of LIN-66 was analysed?

We now specify that LIN-66 isoform C was used for cold shock domain analysis on line 274 of the revised manuscript.

We thank the reviewer for helping us improve the clarity of our manuscript.

May 29, 2024

RE: Life Science Alliance Manuscript #LSA-2024-02673-TR

Prof. Yishi Jin
University of California, San Diego
School of Biological Sciences
La Jolla, CA 92093

Dear Dr. Jin,

Thank you for submitting your revised manuscript entitled "C. elegans LIN-66 mediates EIF-3 G/eIF3-dependent protein translation via a cold-shock domain". We would be happy to publish your paper in Life Science Alliance pending final revisions necessary to meet our formatting guidelines.

- please be sure that the authorship listing and order is correct
- please upload your main manuscript text as an editable doc file
- please add the Twitter handle of your host institute/organization as well as your own or/and one of the authors in our system
- please label the summary as Abstract
- please consult our manuscript preparation guidelines <https://www.life-science-alliance.org/manuscript-prep> and make sure your manuscript sections are in the correct order
- please add a Conflict of Interest statement to your main manuscript text
- there is a callout for Table 2, and there is no Table 2 uploaded; please correct
- please add callouts for Figure S2 and Table S3 to your main manuscript text

A. FINAL FILES:

B. MANUSCRIPT ORGANIZATION AND FORMATTING:

Sincerely,

June 4, 2024

RE: Life Science Alliance Manuscript #LSA-2024-02673-TRR

Prof. Yishi Jin
University of California, San Diego
School of Biological Sciences
La Jolla, CA 92093

Dear Dr. Jin,

Thank you for submitting your Research Article entitled "C. elegans LIN-66 mediates EIF-3 G/eIF3-dependent protein translation via a cold-shock domain". It is a pleasure to let you know that your manuscript is now accepted for publication in Life Science Alliance. Congratulations on this interesting work.

DISTRIBUTION OF MATERIALS:

Again, congratulations on a very nice paper. I hope you found the review process to be constructive and are pleased with how the manuscript was handled editorially. We look forward to future exciting submissions from your lab.

Sincerely,
